# A GRAPH TRANSFORMER FOR SYMBOLIC REGRESSION

## ABSTRACT

Inferring the underlying mathematical expressions from real-world observed data is a central challenge in scientific discovery. Symbolic regression (SR) techniques stand out as a primary method for addressing this challenge, as they explore a function space characterized by interpretable analytical expressions. Recently, transformer-based approaches have gained widespread popularity for solving symbolic regression problems. However, these existing transformer-based models rely on pre-order traversal of expressions as supervision, essentially compressing the information within a computation tree into a token sequence. This compression makes the derived formula highly sensitive to the order of decoded tokens. To address this sensitivity issue, we introduce a novel model architecture called the Graph Transformer (GT), which is purpose-built for directly predicting the tree structure of mathematical formulas. In empirical evaluations, our proposed method demonstrates significant improvements in terms of formula skeleton recovery rates and $R^2$ scores for data fitting when compared to state-of-the-art transformer-based approaches.

## 1 INTRODUCTION

Inferring the underlying mathematical expressions for the real-world observed data is one of the main research problems in the domain of scientific discovery Biggio et al. (2021). More specifically, our objective is to unveil the mathematical formulation by seeking a function denoted as $f$, satisfying the condition that $y_i \approx f(\tilde{x}_i)$ for $M$ observed input-output data pair $\{\tilde{x}_i, y_i\}_{i=1}^M$. Identifying the mathematical expressions governing natural phenomena and technological systems not only provide us with a deeper comprehension of the intrinsic dynamics but also help us to forecast the future evolution of these systems.

There are two primary categories of methods employed to solve this problem. On one hand, machine learning methods like neural networks explore the function $f$ within an extensive range of nonlinear functions by minimizing the loss function across the dataset LeCun et al. (2015). However, the majority of machine learning models are regarded as black-box models, which are more difficult to interpret, and tend to yield poor extrapolation performance because of overfitting Valipour et al. (2021). Conversely, symbolic regression techniques explore in the space of functions characterized by interpretable analytical expressions. The derived mathematical formula can be interpreted and validated in a more intuitive way.

Symbolic regression is considered as NP-hard due to the exponential expansion of the search space with respect to the length of the expression. The presence of numeric constants further exacerbates its difficulty. In this initial stage, genetic programming (GP) Koza (1994) stands out as the most common approach to tackle the symbolic regression problems. GP-based methods iteratively update the candidates of mathematical formulas using the basic operations such as selection, crossover, and mutation. The process of Genetic Programming is discrete optimization, which is known to be computationally expensive and sensitive to hyper-parameters Mundhenk et al. (2021a).

Recent researches made use of the neural network to tackle the aforementioned shortcomings. These neural network-based approaches seek the optimal function by training a parameterized machine learning model through gradient descent, thereby resolving symbolic regression problems within a continuous space Mundhenk et al. (2021a); Martius & Lampert (2016); Sahoo et al. (2018). The well-trained model itself or the output it generates provides the most suitable mathematical function

for fitting a single set of data pairs. Nevertheless, they still treat symbolic regression as an instance-specific challenge, training a new model from scratch on every new dataset.

Motivated by the achievements of large scale pre-training Floridi & Chiriatti (2020), contemporary endeavors in the areas of symbolic regression have shifted their attention toward employing transformer-based models to directly predict mathematical formulas based on input datasets Biggio et al. (2021); Li et al. (2022b); Kamienny et al. (2022). While inferring the underlying mathematical formula from data poses a considerable challenge, the reverse operation, i.e. sampling data points from a formula, only involves evaluation of a mathematical function, and as such is computationally inexpensive. This feature of symbolic regression problems enables the generation of large datasets with minimal computational burden, paving the way for the use of pre-trained models for mathetmatical formula search (as shown in Figure 1).

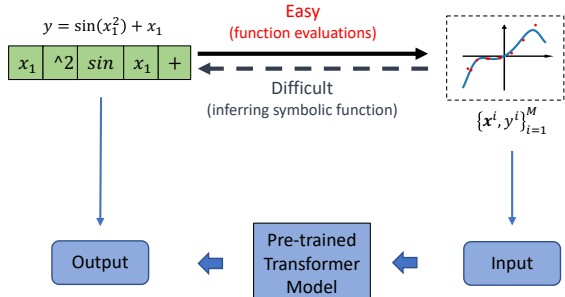

Figure 1: The scheme of using pre-trained models for mathematical formula inference. Despite the primary task is inferring the underlying formula based on observation, we have the ability to easily generate pairs of formulas and observations in the reverse direction.

It should be noted that existing transformer-based models rely on pre-order traversal of expressions as supervision information, compressing the information contained within a computation tree into a sequence of tokens. This approach, however, comes with the drawback of omitting critical details regarding the connections between tokens. Consequently, even a minor alteration in the sequence, as illustrated in Figure 2, such as the mutation of two tokens, can result in a substantial change of the computation tree. In contrast, our work places its emphasis on predicting the connections between tokens, enabling more flexible representations of the formula. For instance, an equation that can only be represented by a unique sequence can be represented by various sequence-adjacency representations, which leads to a higher chance of predicting the correct formula, as shown in Figure 2.

Due to the sensitivity of formula to the order of tokens, we conjecture that a model predicting a tree directly will outperform a model predicting a sequence for symbolic regression tasks. Inspired by established machine learning models designed for graph generation Zhu et al. (2022), we develop a novel model architecture called Graph Transformer (GT). This model is specifically designed to predict the tree structure of mathematical formulas directly. In summary, the primary contributions of this study can be outlined as follows:

- Our innovative framework introduces a paradigm shift by directly generating the computation tree of mathematical formulas, which provides new insights for addressing symbolic regression challenges.

- Our method combines the supervision of the token choice and computation tree skeleton, effectively resolving the problem of decoded formula sensitivity to sequence order.

- In empirical evaluations, our proposed method demonstrates a state-of-art performance compared to recent transformer-based approaches in terms of the recovery rate of formula skeletons and $R^2$ scores for data fitting.

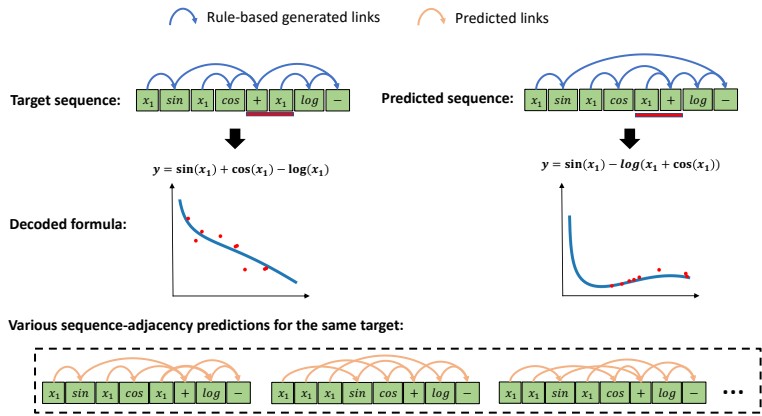

Figure 2: An example to clarify the sensitivity of decoded mathematical formula with respect to token orders and various representation of sequence-adjacency pair for one symbolic equation.

## 2 RELATED WORKS

**Genetic Programming (GP) for symbolic function search.** Traditional approaches to symbolic regression have historically relied on genetic algorithms (GA) Forrest (1993) and, more specifically, genetic programming (GP) Koza (1994). In GP-based symbolic regression, a population of candidate mathematical expressions is iteratively evolved through processes like mutation and crossover. We choose the better candidates based on the fitness function. One of the most well-known GP-based techniques for symbolic regression is Eureqa Dubčáková (2011), a commercial software tool based on the approach proposed in Schmidt & Lipson (2009). While GP methods have showcased the potential of data-driven approaches for function discovery, they face limitations when applied to high-dimensional problems and are highly sensitive to hyperparameters, as highlighted in Mundhenk et al. (2021a). These challenges have spurred the exploration of alternative approaches for symbolic regression, including machine learning-based methods.

**Machine learning for symbolic function search.** Efforts to tackle the challenge of searching for the optimal mathematical formula in a discrete space have indeed encountered significant computational hurdles. Consequently, recent research has shifted its focus towards leveraging machine learning techniques to mitigate the computational cost associated with this task. Some studies replace the activation functions in neural networks with arithmetic operators. This adaptation allows them to harness the power of neural networks for handling high-dimensional data and efficiently scaling with the number of input-output pairs Martius & Lampert (2016); Sahoo et al. (2018). However, when dealing with exponential and logarithmic activation functions, the model training will lead to gradient instability. Another noteworthy approach involves autoregressive models based on reinforcement learning Mundhenk et al. (2021a). In this method, reinforcement learning, specifically utilizing a risk-seeking policy gradient, is employed to train a Recurrent Neural Network (RNN). This RNN is designed to generate a probability distribution over the space of mathematical expressions. Furthermore, there have been endeavors to combine machine learning-based search with Genetic Programming to enhance performance. This integration utilizes the output from the RNN as an initial population for a genetic algorithm Mundhenk et al. (2021b). However, the major limitation of using machine learning for formula searching method is that the network has to be retrained from scratch for each new equation.

**Pre-trained models for symbolic function prediction.** Symbolic mathematics behaves as a language in its own right, where well-formed mathematical expressions are valid "sentences" in this language Valipour et al. (2021). This characteristic has inspired numerous recent studies to tackle symbolic regression problems using sequence prediction models. The initial approach to training large-scale transformer-based models for symbolic regression was established by NeSymReS Biggio et al. (2021). More recently, SymbolicGPT Valipour et al. (2021) trained a GPT Floridi & Chiriatti (2020) model to establish a mapping between pairs of points and symbolic expression strings,

utilizing T-net Qi et al. (2017) as the data encoder. However, these models were limited to predicting only the skeleton of the formula.

To address this limitation, Symformer Vastl et al. (2022) introduced an additional head on the decoder, enabling the model to simultaneously predict the formula skeleton and constants. Another study Kamienny et al. (2022) developed a fully end-to-end model architecture for predicting mathematical formulas, incorporating a novel tokenization method for constant values. While most prior work has focused on addressing the presence of constant values in equations, some studies have aimed to improve the accuracy of formula skeleton prediction. One such approach proposed a novel loss function, combined with contrastive learning Chuang et al. (2020), to provide better supervision for model training, resulting in higher accuracy in predicting formula skeletons. Existing works only consider symbolic regression as a sequence prediction problems, which is fundamentally different from the model architecture we propose in this study.

However, it's important to note that existing works typically treat symbolic regression as a sequence prediction problem. This approach differs fundamentally from the model architecture proposed in our study, which focuses on predicting the computation tree structure of mathematical formulas directly, rather than as a sequence of tokens.

## 3 METHOD

### 3.1 MODEL ARCHITECTURE

The model architecture is visualized in Figure 3. Our machine learning model has an encoder-decoder structure. We first use set transformer Lee et al. (2019) as our model encoder to extract the main features of data pairs $\{\tilde{x}_i, y_i\}_{i=1}^{M}$, denoting the output as points embedding $E_{points}$. We use a trainable hidden embedding (i.e. token), denoted as $T$, to represent the type of each node. The connections between each newly added token and the existing nodes are represented using a binary vector, labeled as $G$, where $G[i] = 1$ indicates the presence of a link from the $i$-th node to the newly added node. Subsequently, our innovative graph decoder generates probabilities for the next token type and the likelihood of link creation based on several inputs: the point embeddings $E_{points}$, the token sequence from previous graph generation steps $[T_0, T_1, ..., T_{n-1}]$, and the connectivity information from previous graph generation steps $[G_0, G_1, ..., G_{n-1}]$.

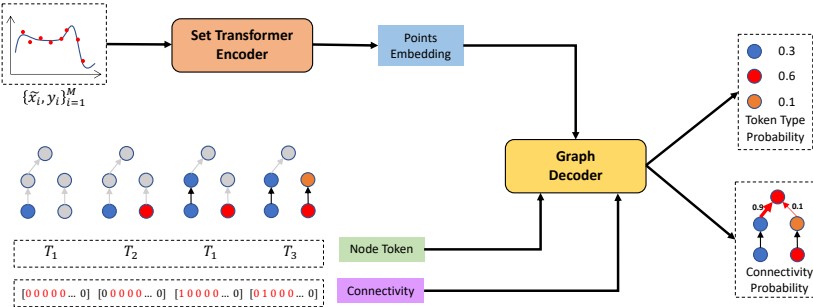

Figure 3: Overall architecture of our model for mathematical formula inferring involves employing a set transformer as the encoder, and introducing an innovative architecture known as the graph decoder. The proposed graph decoder enables the direct prediction of the computation tree for the formula.

To better clarify our proposed graph decoder, we introduce various versions of the attention mechanism utilized in our architecture. The fundamental component in transformer-based models is **single-head Scaled Dot-Product attention module** Vaswani et al. (2017). The output of this attention module is computed by:

$$H = Att(Q, K, V) = softmax(\frac{Q^T K}{\sqrt{d_k}})V, \tag{1}$$

where $Q$, $K$, and $V$ are the query input, key input, and value input of the attention module, $softmax$ is the Softmax operation Jang et al. (2016) for computing the attention scores, and $d_k$ is the dimension size of $Q, K, V$. A more advanced attention module is **multi-head attention module** Vaswani et al. (2017), which aggregates information from different representation subspaces:

$$MH = Multi(Q, K, V) = Concat(H_1, H_2, ..., H_n)W^O, \tag{2}$$

$$H_i = Att(W_i^Q Q, W_i^k K, W_i^v V), \tag{3}$$

where $W_i^Q, W_i^k, W_i^v$ are the linear projection parameters for $Q$, $K$, and $V$, $Concat$ represents concatenation operation and $W^O$ is the linear projection parameters to aggregate the information of different subspace.

The multi-head attention module can be **self-attention** or **cross-attention**, depending on the input sequences. We denote the $S_1$ and $S_2$ as two sequences. In self-attention module, we aggregate the embedding of the sequence $S_1$ based on the itself, while in cross attention we aggregate the embedding $S_1$ based on query computed by the other sequence $S_2$:

$$MH^{self} = Multi(Q = S_1, K = S_1, V = S_1), \tag{4}$$

$$MH^{cross} = Multi(Q = S_2, K = S_1, V = S_1). \tag{5}$$

In the decoder of transformer-based model, we employ a mask in attention module to prevent the model from attending the tokens to be predicted during training, which is called **masked attention module** Vaswani et al. (2017). This makes sure that information should only flow from past to future and not the other way around. Hence, the aggregated embedding of $k$-th token in a sequence is represented as:

$$V_k' = \sum_{i<=k} Att_{k,i} V_i, \tag{6}$$

where $V'$ is the aggregated embedding, $Att$ is the attention scores matrix.

Building upon these different versions of attention modulus, we introduce our novel model architecture, the graph decoder, which generates a graph incrementally, as shown in Figure 4. We use stacked attention layers to predict the next token type and connectivity separately. In each attention layer, we employ a masked multi-head self-attention module to aggregate the input sequence's embedding. Subsequently, we utilize a masked multi-head cross-attention module to connect information between the input sequence and the auxiliary sequence. The auxiliary sequence for token sequence is connectivity sequence, and vice versa. Another multi-head cross-attention module is then employed to aggregate point embedding information into the model, followed by a feed-forward neural network. A normalization layer Ba et al. (2016) will be applied after each attention module and the feed-forward neural network. The output from the stacked attention layers is passed through a linear layer with a Softmax operation to predict token type probabilities or a linear layer with a Sigmoid Han & Moraga (1995) activation function to predict link existence between existing tokens and newly added tokens.

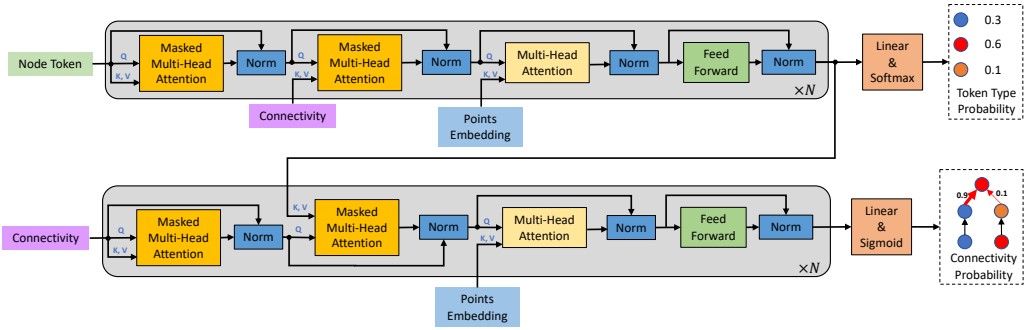

Figure 4: Overall architecture of our proposed graph decoder for generating the computation tree of the mathematical formulas is shown.

## 3.2 MODEL TRAINING AND INFERENCE

We consider the token type prediction as a multi-class classification problem and link prediction as a binary classification problem. Hence our training process involves using cross-entropy (CE) De Boer et al. (2005) loss for symbolic token prediction and negative-log-likelihood (NLL) Bosman & Thierens (2000) loss for link prediction. To train the model effectively, we combine these two terms using a weighted sum denoted as $\mathbf{L}$:

$$\mathbf{L} = \mathbf{L_{CE}} + \lambda \mathbf{L_{NLL}}, \tag{7}$$

where $\lambda$ is a hyperparameter. At the beginning of the training, we set $\lambda$ to zero, and after a few epochs, we gradually increase it using the cosine schedule Loshchilov & Hutter (2016).

During model inference, we use random sampling based on the predicted probability distribution to generate a set of candidate equation skeletons. Then, we implement BFGS Fletcher (2000) to optimize the constant values in the equation skeleton. Finally, we evaluate the performance of each equation candidate on the input data pairs and select the one with the lowest prediction error, thereby determining the best-fitting mathematical equation for the given dataset.

## 4 EXPERIMENTS AND RESULTS

### 4.1 DATASET GENERATION

We introduced a novel framework for generating datasets for tree-generation based symbolic regression models. To create an equation, our framework initiates the process by randomly generating a sequence composed of operands and operators. We adopted a probability distribution for the operators, which aligns with the distribution used in Li et al. (2022a), and employed a uniform distribution for operand sampling. The resulting training dataset comprises approximately 30,000 distinct formula skeletons represented in postfix notation Dabhi & Vij (2011), with placeholders for constants. For each formula skeleton, we generated 20 unique formulas by sampling constant values from a uniform distribution, denoted as $U[1, 5]$, as outlined in Biggio et al. (2021). These sampled constants are then used to fill in the placeholders within the formula skeleton.

The formulas now can be represented as a bunch of sequences of operands and operators, which we refer to as tokens. For each token indexed as $i$, we determined the probability of establishing a link from previous tokens to this token through uniform sampling from the range $U[0, 1]$. Subsequently, we selected the tokens eligible for constructing links, specifically those tokens that have not yet been assigned outgoing links. Finally, we assigned an outgoing link from the token with the highest probability to the $i$-th token if the $i$-th token corresponds to a unary operator, or we assigned two outgoing links if the $i$-th token represents a binary operator. This process governs the connectivity within the sequence of tokens, defining how they relate to each other in the formulation of the mathematical expressions.

Following the previous steps, the entire equation undergoes a simplification process utilizing the rules embedded within the symbolic library SymPy Meurer et al. (2017). We establish input-output pairs $\{\tilde{x}, y\}$ by randomly sampling $\tilde{x}$ from a multivariate standard normal distribution and subsequently calculating the corresponding output value $y$. If the sampling of $\tilde{x}$ results in non-finite values (such as NaN or $\pm\infty$), we repeat the sampling process until a valid value is obtained. If the sampling process exceeds a time limit of 30 seconds without producing a valid result, we discard the expression associated with that particular set of inputs. This ensures that our dataset comprises valid and meaningful input-output pairs for training and evaluation.

### 4.2 BASELINE MODELS

We employed the Adam optimizer Zhang (2018) and distributed the training across 4 NVIDIA A100 GPUs for computational efficiency. The learning rate was set to 0.0001, and we used a batch size of 64 to update model parameters in each training iteration. For evaluating the performance of our model, we conducted comparisons with several state-of-the-art baseline methods:

- Genetic Programming (GP) Koza (1994): Standard GP-based symbolic regression based on the open-source Python library gplearn Stephens (2019).

- Deep Symbolic Optimization (DSO) Mundhenk et al. (2021a): A symbolic regression method based on RNN and reinforcement learning search strategy.
- Neural Symbolic Regression that Scales (NeSymReS) Biggio et al. (2021): the first transformer-based symbolic regression model on the large training data.
- Transformer-based model for symbolic regression via Joint Supervised Learning (T-JS) Li et al. (2022b): transformer-based symbolic regression model with the training loss combined with contrastive learning.
- End-to-end symbolic regression with transformers (E2E-SR) Kamienny et al. (2022): A recently proposed novel model architecture for end-to-end symbolic equation prediction.

During the training phase, we ensured a fair comparison among different transformer-based model architectures by utilizing the same dataset for their training. Additionally, all of these models shared a common token library for the decoder. The model-specific parameters were kept the same as the implementation of original papers.

Our model used 8 decoder layers and 4 encoder layers. The hidden dimension of the embedding is 512. The maximum size of the decoded sequence of our model is 100. We trained each model for 10 epochs and chose the one of highest performance for testing.

## 4.3 METRICS

We utilize two criteria to evaluate the model performances: the skeleton recovery rate and the coefficient of determination $R^2$ values Glantz et al. (2001). The skeleton recovery rate is evaluated based on the predicted token sequence and the corresponding adjacency matrix of the computation tree, measuring the model's ability to accurately reconstruct the underlying structure. The coefficient of determination $R^2$ is calculated using the formula:

$$R^2(y, \hat{y}) = 1 - \frac{\sum_{i=1}^{K}(y_i - \hat{y}_i)^2}{\sum_{i=1}^{K}(y_i - \bar{y})^2}, \tag{8}$$

where $y_i$ and $\hat{y}_i$ are the ground-truth and predicted values for point $\tilde{x}_i$, respectively, $\bar{y}$ is the average of $y_i$ over all the points, and $K$ is the number of test points. A higher $R^2$ value indicates a superior predictive performance. When $R^2$ is greater than 0, it means that the prediction is better than merely predicting the average value, suggesting that the model provides better performance than a basic average prediction.

## 4.4 IN-DOMAIN PERFORMANCE

Our initial evaluation focuses on the model's ability to accurately detect the exact mathematical formulas within our self-synthesized dataset. In this assessment, we exclusively compare our method with other pre-trained-based models that predict the formula directly. As depicted in Table 3, our method demonstrates a higher recovery rate of formula skeletons compared to all the pre-trained-based methods.

Table 1: Recovery rate of equation skeletons comparison between multiple pre-trained models on the self-synthesized benchmark are shown.

| Model | Input Dimension | | |
|---|---|---|---|
| | $x \in R$ | $x \in R^2$ | $x \in R^3$ |
| NeSymReS | 62.4% | 57.2% | 47.2% |
| T-JS | 74.5% | 66.3% | 55.1% |
| E2E-SR | 68.9% | 60.8% | 51.4% |
| GT | **81.9%** | **70.2%** | **61.4%** |

We further assessed the model's performance in terms of data fitting on our self-synthesized dataset. As evident from the results in Table 3, our method surpasses all baseline methods with respect to the average $R^2$ score. The $R^2$ scores for one-dimensional data fitting exceed those for higher-dimensional data fitting, indicating the increased difficulty of fitting higher-dimensional data on average. We also observed that there is a positive correlation between the skeleton recovery rate and the $R^2$ scores, with higher skeleton recovery rates corresponding to higher $R^2$ scores.

Table 2: The $R^2$ scores of different methods for data fitting on the self-synthesized benchmark are shown.

| Model | Input Dimension | | |
|---|---|---|---|
| | $x \in R$ | $x \in R^2$ | $x \in R^3$ |
| GP | 0.9032 | 0.8827 | 0.8495 |
| DSO | 0.9767 | 0.9263 | 0.8954 |
| NeSymReS | 0.9454 | 0.9029 | 0.8693 |
| T-JS | 0.9879 | 0.9531 | 0.9117 |
| E2E-SR | 0.9802 | 0.9623 | 0.9318 |
| GT | **0.9932** | **0.9704** | **0.9583** |

## 4.5 OUT-OF-DOMAIN PERFORMANCE

We conducted a comprehensive evaluation of model performances in uncovering dynamics from observed datasets using the recently introduced SRbenchmark dataset La Cava et al. (2021). Our specific focus was on "black-box" problems, which involve a combination of real-world and noisy synthetic datasets. For each "black-box" dataset, we created 50 dataset samples by randomly sampling features from the input and identify the function that best captured the relationship between a subset of features and the target value. The results, as illustrated in Figure 3, consistently demonstrate that our model outperformed all existing baseline models across various "black-box" datasets. These findings underscore the effectiveness of our approach in addressing complex, real-world data-driven challenges.

Table 3: The $R^2$ scores of different methods for data fitting on the SRbenchmark are shown.

| Model | Input Dimension | | |
|---|---|---|---|
| | $x \in R$ | $x \in R^2$ | $x \in R^3$ |
| GP | 0.4493 | 0.4279 | 0.4098 |
| DSO | 0.4682 | 0.4398 | 0.4128 |
| NeSymReS | 0.4754 | 0.4432 | 0.4045 |
| T-JS | 0.4931 | 0.4789 | 0.4591 |
| E2E-SR | 0.5292 | 0.5074 | 0.4813 |
| GT | **0.5583** | **0.5304** | **0.5211** |

We also conducted a comparison focusing on two aspects: the complexity of decoded formulas and the inference time. As demonstrated in Figure 5a, our model generated less complex functions compared to other baseline models. This reduction in complexity enhances the interpretability of the mathematical formulas generated by our model. However, it can be seen that our model required more time for formula decoding due to its more complicated architecture. Although the decoding time is longer compared to other baseline models, it remains within a reasonable range, ensuring practical usability.

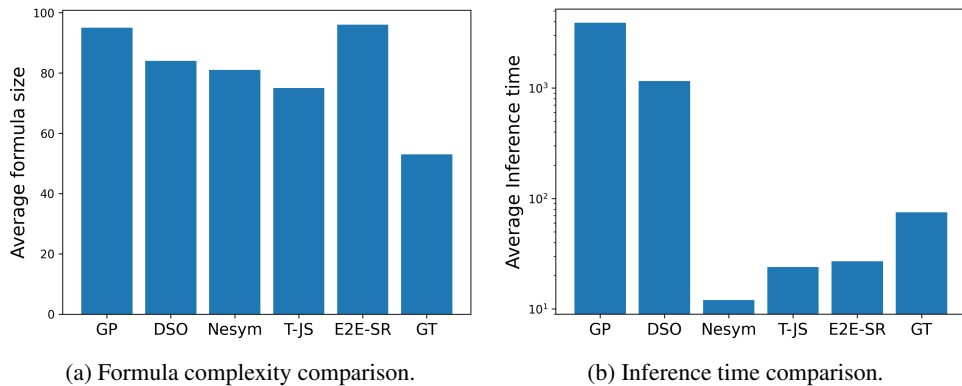

(a) Formula complexity comparison.  (b) Inference time comparison.

Figure 5: The complexity (number of tokens) of the decoded formulas and average inference time for the SRbenchmark dataset of different methods are shown.

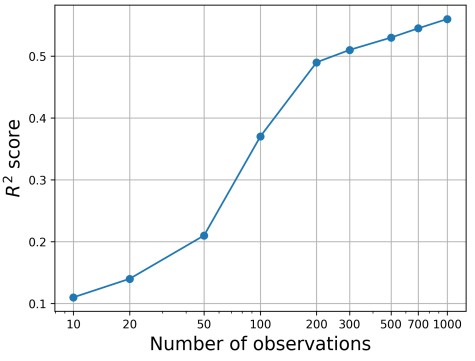 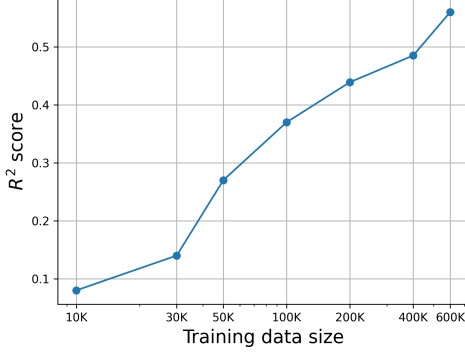

(a) Model performance versus the number of input observation.

(b) Model performance versus the training dataset size.

Figure 6: Our model performance versus the number of observation for each formula (left) and versus the size of the training dataset (right) are shown.

We further conducted an investigation into the impact of the number of input observations and training data size on model performance. As demonstrated in Figure 6a, we observed that the model's performance improved as the number of input observations increased. However, beyond a certain threshold (around 200 observations), the number of observations had a limited influence on model performance.

Similar conclusions were drawn when assessing the effect of training data size on model performance, as shown in Figure 6b. The model's performance benefited from a larger dataset, and we did not observe a plateau in performance up to a training data size of 600K equations. This suggested that further improvements in model performance may be achievable with even larger datasets, indicating the potential for continued improvement in model capabilities.

## 5 CONCLUSION

In this study, we propose a novel framework to directly predict the computation tree structure of mathematical formulas directly. Using this framework, we eliminates the need for information compression of a computation tree into a sequence. The experimental results demonstrate that our model can achieve state-of-the-art performance in the task of inferring mathematical formulas. This work represents a significant advancement in the field of symbolic regression and opens up a new range of machine learning methods in this domain. We anticipate that the methods presented here will serve as a valuable toolbox for the development of more novel model architectures for computation tree prediction.

**Limitation** This work presents several limitations. Firstly, the current model architecture has been tested primarily on low-dimensional symbolic regression problems. Adapting the framework to high-dimensional scenarios represents an interesting and challenging future direction, potentially requiring substantial changes in data generation protocols and model architectures. Secondly, the current model architecture relies on an attention mechanism with computational complexity of $O(L^2)$. Combining this with graph neural networks could potentially lead to efficiency increment in solving symbolic regression tasks. Lastly, the current framework is designed to predict the skeleton of mathematical formulas, and extending it to achieve end-to-end formula prediction, including constant values, would be an intriguing topic for future research.

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
