# OpenReview forum: "A graph transformer for symbolic regression"
_ICLR.cc/2024/Conference — ICLR 2024 Conference Withdrawn Submission_

### Official Review · Reviewer_YZQC · 2023-10-15

**Soundness:** 3 good
**Presentation:** 1 poor
**Contribution:** 2 fair
**Rating:** 3
**Confidence:** 4

**Summary:**

This work considers the problem of performing symbolic regression (SR) by training a model on a synthetic dataset of paired input/output samples and symbolic functions. The authors point out that existing approaches for this problem have relied on pre-order-traversal representations of the symbolic functions to convert them into token sequences, and argue that this representation has drawbacks because the "connections" between tokens are not explicitly fed into the model. They then propose an alternative model ("graph transformer") which predicts explicit edges connecting operations to their arguments, instead of using a pre-order traversal. The authors demonstrate that their model outperforms baseline SR methods on synthetic data and on a subset of the SRbenchmark dataset.

**Strengths:**

**[S1]** The overall idea of using a tree-based output representation during symbolic regression makes sense and seems like an appropriate representation for the task. I'm not aware of edge-prediction-based models being previously applied to this task (although I am not that familiar with the state of the art in symbolic regression).

**[S2]** The high-level overview figures clearly explain the problem setting (Fig 1), order-dependence of the traversal (Fig 2), and high-level structure of the proposed architecture (Fig 3 and 4).

**[S3]** The empirical results show that their method does quite well relative to existing baselines. The authors also include separate results based on the dimensionality of the input, number of observations, and training dataset size.

**Weaknesses:**

**[W1]** This work does not acknowledge or cite the substantial body of existing work on autoregressive graph generation. The problem of generating graphs in general, and trees of operators in particular, has been considered in many previous works, and the architecture proposed here seems to have only minor differences from previously proposed models.

- ["Learning Deep Generative Models of Graphs" (Li et al. 2018)](https://arxiv.org/abs/1803.03324) describes a general framework for generating graphs by sequentially adding nodes, predicting their types, and adding edges.
- ["Order Matters: Probabilistic Modeling of Node Sequence for Graph Generation" (Chen et al. 2021)](https://arxiv.org/abs/2106.06189) discusses the impact of node order for graph generation models.
- ["Generative code modeling with graphs" (Brockschmidt et al. 2018)](https://arxiv.org/abs/1805.08490) uses a generative graph model to build source code expressions, which are very similar to the symbolic regression expressions considered here.
- ["Efficient Graph Generation with Graph Recurrent Attention Networks" (Liao et al. 2020)](https://arxiv.org/abs/1910.00760) proposes an attention-based generative model for graphs,
- From a fairly cursory search also I found multiple works proposing similar domain-specific transformer-based generative models of graphs for other domains, e.g. [Cofala et al. (2021)](https://www.semanticscholar.org/paper/Transformers-for-Molecular-Graph-Generation-Cofala-Kramer/30161db106745baa85bf2e2493a81b96ad8ad6db), [Zhong et al. (2022)](https://dl.acm.org/doi/abs/10.1145/3503161.3548424), [Belli et al. (2019)](https://arxiv.org/abs/1910.14388)
- [Guo et al. 2022](https://arxiv.org/abs/2007.06686) gives an overview of many of the previous works in this space and of the general types of architecture that have been considered before. (This might be a good starting point for performing a more thorough literature review of related prior work in this area.)

These are just a representative sample. I think the authors of this work should perform a more thorough literature search and give appropriate discussion of this subfield in their related work section.

**[W2]** The proposed approach does not seem particularly novel to me as a whole; it seems to me like a combination of a well-studied technique and a well-studied problem area in a slightly new way. The authors also inappropriately praise the novelty and significance of their own approach many times, e.g. "Our innovative framework introduces a paradigm shift ...", "our innovative graph decoder ...", "This work represents a significant advancement in the field ...". In general I think this kind of claim is usually out of place in a paper submission, as it is up to the community to judge the significance of a work.

**[W3]** The submission is missing many important details that would be necessary to fully understand or reproduce the results, or to implement the architecture itself.

- The paper makes multiple mentions of token types and node types, but what these are is never clearly explained. I assume they must be the operations and leaves of the symbolic regression tree?
- Similarly, the paper doesn't clearly explain the rules governing connections, although it can be partially reverse-engineered from Figure 2. (The "sequence-adjacency pair" terminology is also never defined.)
- I couldn't find any discussion of how the order of the tokens is selected. The authors note that there are multiple possible orderings for the same target, but which ordering is actually used? Since the model generates in a sequence, there must still be some ordering that the model is trained to produce.
- The description of the dataset generation is confusing and leaves out a lot of details. Additionally, I didn't understand how the formula skeleton and links are chosen separately, since it seems to me that a formula skeleton could only have a single set of possible links. (See my questions below.)
- The paper also doesn't seem to show any examples from the proposed synthetic dataset, or any outputs of their model.

(On the other hand, I think the paper goes into an unnecessary amount of detail about the basics of self-attention and cross-attention, which are described in previous work and which many readers are likely already familiar with. In my opinion it would make more sense to focus on the novel parts of the architecture instead.)

**[W4]** The paper has a number of minor presentation and style issues.

- The name "Graph Transformer" is very generic, and this term has already been used in many previous works to refer to transformer models applied to graphs (e.g. [Rampášek et al. (2022)](https://arxiv.org/abs/2205.12454), [Kim et al. (2022)](https://arxiv.org/abs/2207.02505), [Cai et al. (2019)](https://arxiv.org/abs/1911.07470)). Also, the model proposed in this work seems only capable of generating trees, not general graphs. I'd encourage the authors to use a more specific name.
- Many citations appear to use `\citet` when `\citep` would be more appropriate.
- Figure 2 seems to show a post-order traversal, but is described as a pre-order traversal in the main text.
- Section 4.4 refers to "Table 3" but this appears to be the wrong table.

**Questions:**

In section 1, what does "this initial stage" refer to?

Section 1 mentions that the pre-order traversal representation "comes with the drawback of omitting critical details regarding the connections between tokens". It seems to me that that information isn't missing, it's still encoded in the traversal, right? Could you be more specific about what you are arguing here?

Figure 2 shows an example of swapping the order of two tokens. However, this seems unrealistic; is there a reason we would expect a model to swap two tokens during generation? (Also, does your method work with every possible token swap? It seems like certain token swaps would still be impossible to encode with your method, since links all point in the same direction.)

Section 4.1 refers to the example generation process as "a novel framework", but it's not obvious to me what's novel about it; it seems fairly similar to the method used in prior work (e.g. in Kamienny et al.). What is the novelty here? Does your strategy fix any problem with the previous one?

Also, in Section 4.1, I don't understand how the link generation works. From the first paragraph it sounds like you are generating distinct formula skeletons in postfix notation. But given the formula in postfix notation, all of the connections are implied by the postfix structure, are they not? So how are you randomly choosing links in the second step, if the arguments to the operands are already determined by the formula skeleton?

---

### Official Review · Reviewer_dKpQ · 2023-10-18

**Soundness:** 3 good
**Presentation:** 2 fair
**Contribution:** 2 fair
**Rating:** 5
**Confidence:** 4

**Summary:**

This paper introduces a graph transformer architecture to address the problem of predicting a mathematical expression underlying a set of observation, known as symbolic regression (SR).

**Strengths:**

- Architecture: the graph decoder presented here is an interesting architecture which indeed could be better adapted to SR (but probably not a game-changer as I discuss below)
- Presentation: the paper is rather well presented and well written

**Weaknesses:**

- Experimental validation: The authors only present results on the SRBench black-box problems. These are arguably the least interesting from a SR point-of-view : the Feynman problems are far more adapted to SR. This looks like a voluntary omission, and I require the authors to present these results during rebuttal, otherwise I cannot recommend acceptance.
- Lack of details: the authors omit a few details in the implementation, as discussed below.
- Potential impact: I do not think this approach is well-suited to advance the state-of-the-art in SR. Their proposed architecture addresses the problem of the expression syntax, which is an interesting problem but likely not a bottleneck for SR (more details below), meanwhile as acknowledged in the limitation section their approach only predicts skeletons, which on the other hand is a strong bottleneck in terms of performance as demonstrated by Kamienny et al [1]. However this is my personal opinion and it will not come into play in my general appreciation of this paper.

[1] Kamienny, Pierre-Alexandre, et al. "End-to-end symbolic regression with transformers." Advances in Neural Information Processing Systems 35 (2022): 10269-10281.

**Questions:**

- While I do agree that a graph-based encoding of expressions is better suited than a sequence-based encoding, I do not believe that syntax is not the main bottleneck for symbolic regression with Transformers, in the same way that syntax is the not the bottleneck for reasoning in LLMs. The mathematical challenge of predicting all the coefficients and the skeleton of the formula from limited information is the main bottleneck, and I don’t this using Graph Transformers rather than seq2seq Transformers fundamentally addresses it. It is unclear to me how the inputs (x,y) and the connectivity vectors [0 1 0 …] are embedded as single tokens. The authors should add a paragraph on this.
- The authors do not specify on which input dimensionalities the model is trained, and whether they use different models for each dimension or the same one.
- I don’t understand the skeleton recovery metric. How exactly is it calculated for the baseline models which do not generate graphs ?
- The authors should add a sentence acknowledging that testing on the self-synthesized dataset favours their own model compared to others.

---

### Official Review · Reviewer_SCmX · 2023-10-29

**Soundness:** 3 good
**Presentation:** 2 fair
**Contribution:** 3 good
**Rating:** 5
**Confidence:** 4

**Summary:**

Symbolic regression is an effective way to infer the underlying mathematical expressions based on observed data. Mathematical expression can be naturally expressed as a tree-based computational graph. The paper attempts to directly recover the tree structure based on a graph-transformer model rather than generate a pre-order traversal of the computation graph since order recover is a very sensitive task. Experimental study demonstrated that the proposed method is superior than existing transformer based methods.

**Strengths:**

1.The paper view symbolic regression as a graph generation problem which directly recovers the tree structure of computational expressions and overcomes the order sensitive issue in pre-order traversal sequence generation.
2.Experiments over in-domain/out-domain data show superior performance comparing to some SOTA methods.

**Weaknesses:**

1.The proposed method is only evaluated on low-dimensional symbolic regression problem, and cannot directly handle constant values.
2.The introduction of the generation procedure should be refined, especially the structure generation part (please see question part)

**Questions:**

1.For decoding part, how many different token types are used in the paper? Is the token type decoded greedily? The candidate skeletons is generated based on random sampling. What is the sample size? Did the proposed method make use of beam search to control the sample quality? How to determine the termination of the generation procedure？
2.The computation graph of mathematical expression is a tree-structure, therefore, did the decoding procedure make use of such prior information? For example, circle detection.
3.What is the range of lambda in Eq. (7)? What is the motivation of setting it as zero at first and then increasing with cosine schedule?
4.In section 3.1, it is better to introduce the attention module based on the problem definition. For example, in decoder phase, what does query, key (such as points embedding), value mean? This should be consistent to Fig. 4.
5.How to calculate the inference time of GT? Did this time also include BFGS for constant value optimization?

---

### Official Review · Reviewer_mTB4 · 2023-10-31

**Soundness:** 2 fair
**Presentation:** 2 fair
**Contribution:** 2 fair
**Rating:** 3
**Confidence:** 4

**Summary:**

This paper proposes a symbolic regression method named Graph Transformer (GT). GT is developed as a direct estimator of the tree structure of mathematical formulas by estimating the adjacency between the token nodes. GT is compared to other methods using synthetic data and a part of SRBench, and it is reported that GT shows better accuracy.

**Strengths:**

- The proposal of a Transformer-based decoder that generates graphs autoregressively by simultaneously predicting node tokens and their adjacencies, and its application to symbolic regression is a novel approach.
- Experimental results show that GT achieves better accuracy than recent transformer-based and other methods.

**Weaknesses:**

- Although the author claim that the difference between GT and previous studies is that it focuses on directly predicting the computation tree, for example, (Kamienny et al., 2022) estimates the tree structure of formulas as series data using reverse Poland notation. Mathematical expressions in such notation are basically equivalent to computation tree; it is not clear how the proposed method is fundamentally different in predicting comptation tree.
- The proposed method, like other Transformer-based methods, generates new tokens in an autoregressive manner and computes adjacencies to existing tokens.
  - It is said that cross entropy is used to learn to add these tokens, but the order in which tokens are added should not be uniquely determined. CE loss may not be an appropriate loss function.
  - Furthermore, the reviewer guesses that negative-log-liklihood loss to learn to estimate the link between additional tokens and existing tokens is also based on a fixed order from the supervision. If so, even if the tokens are correctly added in a different order than the supervision, and even if the link is correctly estimated, a loss will be incurred.
- Although GT proposes to decode the graph structure directly, the experimental result is not a comparison of superiority or inferiority by decoder, because the structure of the encoder is also different from other existing Transformer-based methods. The consistency between what is shown in the experiments and the claims in this paper is not guaranteed unless the authors at least conduct an ablation study using the same encoder as the proposed method, but using a series prediction-based decoder, for example.

**Questions:**

- The reviewer expects the authors to respond to the points listed in Weaknesses.
- All comparisons with existing methods are made on each input dimension. This is a somewhat peculiar comparison in light of related studies, and the reviewer would like to ask the authors what their motivation is for adopting this comparison.
- In addition to the black-box problems, SRBench contains 130 problems with correct answers from the Feynman Symbolic Regression Database and the ODE-Strogatz Database. The reviewers would like to know why the paper did not use these data to compare the proposed method with existing studies.